# On the Fly: Tritrophic Associations of Bats, Bat Flies, and Fungi

**DOI:** 10.3390/jof6040361

**Published:** 2020-12-12

**Authors:** Michiel D. de Groot, Iris Dumolein, Thomas Hiller, Attila D. Sándor, Tamara Szentiványi, Menno Schilthuizen, M. Catherine Aime, Annemieke Verbeken, Danny Haelewaters

**Affiliations:** 1Research Group Evolutionary Ecology, Naturalis Biodiversity Center, Darwinweg 2, 2333CR Leiden, The Netherlands; menno.schilthuizen@naturalis.nl; 2Research Group Mycology, Department of Biology, Ghent University, K.L. Ledeganckstraat 35, 9000 Ghent, Belgium; iris.dumolein@ugent.be (I.D.); mieke.verbeken@ugent.be (A.V.); danny.haelewaters@gmail.com (D.H.); 3Ecology of Tropical Agricultural Systems, University of Hohenheim, Garbenstrasse 13, 70599 Stuttgart, Germany; thomas.hiller@uni-hohenheim.de; 4Institute of Evolutionary Ecology and Conservation Genomics, University of Ulm, Albert-Einstein-Allee 11, 89081 Ulm, Germany; 5Smithsonian Tropical Research Institute, Apartado 0843-03092, Balboa, Panama; 6Department of Parasitology and Parasitic Diseases, University of Agricultural Sciences and Veterinary Medicine, Calea Mănăștur 3-5, 400372 Cluj-Napoca, Romania; adsandor@gmail.com; 7Department of Parasitology and Zoology, University of Veterinary Medicine, István u. 2, 1078 Budapest, Hungary; 8Pathogen and Microbiome Institute, Northern Arizona University, 1395 S. Knoles Drive, Flagstaff, AZ 86011, USA; tamaraszentivanyi@gmail.com; 9Department of Botany and Plant Pathology, Purdue University, 915 W. State Street, West Lafayette, IN 47906, USA; maime@purdue.edu

**Keywords:** bat flies, Chiroptera, food webs, host specificity, hyperparasites, Laboulbeniales, parasites

## Abstract

Parasitism is one of the most diverse and abundant modes of life, and of great ecological and evolutionary importance. Notwithstanding, large groups of parasites remain relatively understudied. One particularly unique form of parasitism is hyperparasitism, where a parasite is parasitized itself. Bats (Chiroptera) may be parasitized by bat flies (Diptera: Hippoboscoidea), obligate blood-sucking parasites, which in turn may be parasitized by hyperparasitic fungi, Laboulbeniales (Ascomycota: Laboulbeniomycetes). In this study, we present the global tritrophic associations among species within these groups and analyze their host specificity patterns. Bats, bat flies, and Laboulbeniales fungi are shown to form complex networks, and sixteen new associations are revealed. Bat flies are highly host-specific compared to Laboulbeniales. We discuss possible future avenues of study with regard to the dispersal of the fungi, abiotic factors influencing the parasite prevalence, and ecomorphology of the bat fly parasites.

## 1. Introduction

The world today is undergoing fast anthropogenic changes. Five major changes pose a direct threat to biodiversity and ecosystem functioning: climate change, habitat alteration, exploitation, pollution, and invasive species [1]. While scientists try to halt biodiversity loss through conservation strategies, they often seem to neglect considering parasites [2,3]. Parasitism is one of the most diverse modes of life [4,5]. Parasites are thought to have evolved as often as, if not more often than, other types of organisms [6,7]. Parasites are important drivers of adaptive change [8]. They are known to alter food webs either indirectly through enhancing their transmission rate by altering life-history traits of their hosts, or directly by changing the food chain length, number of links, and energy flows, with possibly as many as 75% of links in food webs involving a parasitic species [9,10]. Finally, parasites play important roles as bioregulators of population dynamics.

Whereas they are prevalent and vital for their ecosystem, parasites are vastly understudied and the effects of parasites on their hosts and vice versa are often still unclear [3,11]. Parasites face the same and perhaps even larger threats compared to other organisms due to their dependence on other species for their survival [5]. Obligate parasites, those that cannot survive without a suitable host, are often co-endangered once the ecosystem and natural relations are out of balance; when a host faces endangerment or extinction, as a consequence, its parasite(s) will too. Ultimately, when the host goes extinct, this will lead to co-extinction of its parasite(s), resulting in a cascade reaction of secondary extinctions [12]. As a result, the loss of parasite biodiversity contributes a significant amount to the current sixth mass extinction [5]. This calls for a need to include parasites in integrative conservation strategies.

An important determinant of loss of parasite biodiversity is host specificity. It is known that in some cases, when parasites are less host-specific, they can save themselves from co-extinction by shifting to alternative hosts [13,14]. Host specificity of parasites is a key factor determining their geographical range [15,16], their ability to colonize and survive on a new host [17], and the likelihood of host-parasite co-extinction risk [18]. Host specificity can be measured at different levels: (i) structural specificity, the distribution of parasite populations across hosts and the relative frequency specific parasites exploit specific hosts, (ii) phylogenetic specificity, a measure of phylogenetic relatedness among hosts that a parasite prefers or the phylogenetic relatedness of parasites that prefer a specific host, and (iii) geographical specificity, a measure of how parasites distribute themselves across hosts on different spatial scales [19].

One particular type of parasitism is hyperparasitism (Figure 1). Hyperparasitism is an interaction where a parasite itself is infected with another parasite [20]. Although it might appear “a risky lifestyle” at first glance, hyperparasitism is thought to be a common phenomenon in nature [20,21,22]. Hyperparasites can play a role in altering the population sizes of their hosts. Since these hosts are parasites themselves, the hyperparasites may indirectly affect the primary host population sizes, too [23]. As such, tritrophic interactions can alter the patterns of energy flow in food webs and have an impact on the overall biodiversity as well as population dynamics. One example of obligate hyperparasitism, where one parasite is parasitized by a secondary parasite that *on its own* cannot parasitize an uninfected primary host [24], is presented by the associations found among bats (Mammalia: Chiroptera), bat flies (Diptera: Hippoboscoidea), and Laboulbeniales fungi (Ascomycota: Laboulbeniomycetes). 

Bats are nocturnal animals with a long life-span, slow reproduction rates, a high metabolism, improved mobility, and variable in sociality—living either solitary or in large colonies. The clade is the second-largest order within Mammalia, with about 1426 described species [25]. They inhabit different roosting structures from ephemeral leaf tents to caves and mines [26]. Bats are keystone species with many ecosystem functions, such as seed dispersal, pollination, insect population control, and suppression of pest-associated fungal growth and mycotoxin in corn [27,28]. Sixteen percent of bat species are listed in threatened categories: critically endangered (CE), endangered (EN), and vulnerable (VU), whereas seven percent are near threatened (NT) [29]. These numbers indicate that bat populations around the world are facing serious threats [30]. The ecology and biology of bats contribute to the fact that they are parasitized by different lineages of organisms, including bugs (Hemiptera), earwigs (Dermaptera) (but see [31]), mites and ticks (Acari), fleas (Siphonaptera), and, most conspicuously, the bat flies (Diptera) [32,33].

Bat flies are obligate, largely host-specific, bloodsucking ectoparasites of bats [34]. They are traditionally divided into two families: the Nycteribiidae, which are most diversified in the Eastern Hemisphere, and the non-monophyletic Streblidae, which are most species-rich in the Western Hemisphere [35,36]. Nycteribiid bat flies are wingless, dorsoventrally flattened and have dorsally inserted legs; they vary in size from 1 to 5 mm [37]. Streblidae, on the other hand, are highly variable in their external morphology; they might be laterally compressed, dorsoventrally flattened, or uncompressed [38,39,40,41]. Most streblid species have functional wings. Bat flies reproduce by adenotrophic viviparity; a fertilized egg hatches inside the female fly, followed by three larval stages that are carried inside the female, nourished by an intrauterine accessory or “milk” gland. The third instar larva is deposited onto a suitable substrate in the bat roosting environment, such as a cave wall. After a developmental process of 3–4 weeks, the imago emerges in search of a bat host [39]. Diverse parasitic organisms have been reported from bat flies [42], including microscopic fungi of the ascomycete order Laboulbeniales.

Laboulbeniales are obligate biotrophic ectoparasites, producing two-celled ascospores from which a 3-dimensional thallus composed of true parenchyma develops through successive divisions in multiple planes [43]. The entire life cycle of Laboulbeniales takes place on the exoskeleton of the host, and free-living asexual stages are thus far unknown. Laboulbeniales are often host-specific [44,45]. Transmission of ascospores largely happens during host activities, such as mating and social grooming; indirect transmission is less often, due in part to the low survivability of ascospores in the environment (<7 days) [46]. Laboulbeniales exhibit a large morphological and structural diversity, as well as a wide arthropod host range, which place them among the most diverse groups of parasitic fungi with ~2325 described species in 145 genera [47,48,49]. Only four genera are found on bat flies: *Arthrorhynchus*, *Dimeromyces*, *Gloeandromyces*, and *Nycteromyces* [33,50]. Based on the little data available in the literature, Laboulbeniales seem to prefer female bat flies. This has been attributed to the longer life span of females, their larger size, and their fat reserves during pregnancy [51].

Here, we investigate the tripartite interaction of bats–bat flies–Laboulbeniales and measure the structural specificity of the parasites. As previously described, the parasitic organisms in this multitrophic association face the same threats as a result of co-endangerment as their hosts. In order to get a full picture of how this tripartite association may react to current anthropogenic changes, it is important to fully disentangle the interactions among different partners. We hope that our work can aid in developing better conservation methods, both for bats and their parasites.

## 2. Materials and Methods

Bats were captured between 1993 and 2018 in several countries in the Neotropics and Europe using mist nets or harp traps located at drinking, foraging, and swarming sites, as well as over trails that presumably functioned as flight pathways for bats. Bats were identified to species level [52,53,54,55]. Bat flies were collected using forceps, by hand, or with the help of a Fair Isle apparatus [56]. Bats were released at the vicinity of the capture site immediately after processing. Long-term storage of bat flies was in 70–96% ethanol in separate vials (one vial per bat host). Bat flies were identified using identification keys and complementary publications [57,58,59,60,61,62,63,64,65,66,67,68,69,70,71]. 

Bat flies were screened for the presence of Laboulbeniales thalli at 40–50× magnification. Thalli were removed from their host at the attachment region and slide-mounted [22,72] for identification to species level [51,72,73,74,75]. The results of this work have been incorporated into a database currently holding 11,936 bat flies with associated metadata: bat fly species and sex; bat host identification, sex, age, and reproduction status; presence/absence of Laboulbeniales, the position of infection, and fungus identification; geographic location and collecting date. Parts of this dataset have been previously published [41,51,72,76,77,78,79]. A complete list of locations where infected bat flies were collected can be found in Appendix A. 

For the construction of an association web, a subset of bat flies—those infected by Laboulbeniales (Nycteribiidae: *n* = 45; Streblidae: *n* = 287; total *n* = 332)—and their associated bats were used. In a few cases, the identification of the Laboulbeniales could not be determined to species level. These were all juvenile thalli of *Gloeandromyces*, which do not exhibit sufficient morphological features for accurate identification. These associations were grouped within “*Gloeandromyces* spp.” In addition, some thalli of Laboulbeniales did not fit any known descriptions and thus may represent undescribed species. This was the case for thalli on three Panamanian bat flies identified as *Strebla galindoi* (collected from *Tonatia saurophila*, the Stripe-headed Round-eared Bat), listed in the association web as *Gloeandromyces* sp. nov. Finally, three Mexican specimens of bat fly represented an undescribed species of *Trichobius* (collected from *Choeronycteris mexicana*, the Mexican Long-Tongued Bat); they are reported in the association web as *Trichobius* sp. nov. Association webs were generated using the R language and environment for statistical computing [80], with the ‘bipartite’ package [81]. Species were ordered by region (Central and South America versus Europe). 

Structural host specificity was described using two metrics: *H*_2_’ and *d*’*_i_* (*sensu* [82]). The *H*_2_’ index is a community-level measure of host specificity, ranging from 0 (completely generalist community) to 1 (completely specialist community). The value of the *H*_2_’ index increases as the species distribution deviates more from a null model where all species interact with other species according to their proportion of their observed frequency [82]. The *d*’*_i_* index is a metric for the specificity of each individual node—or, species-level host specificity. Like the *H*_2_’ index, its values range from 0 (most generalist species) to 1 (most specialist species). The value of *d*’*_i_* increases as a species *i* is observed to interact in a way that deviates from the expected distribution from a null model where all species interact in proportion to their observed numbers. These host specificity metrics were calculated using the “bipartite” package [81], with functions “H2fun” and “dfun”. 

In aiming to explain the presence versus absence of Laboulbeniales on certain bat flies, we hypothesized that parasitism by Laboulbeniales may be affected by bat fly ecomorphology. Streblid bat flies can be divided into three groups: wing crawlers, which feed on the membranes of the bat’s wings; fur runners, which have very long posterior legs and run over the top of the fur when disturbed; and fur swimmers, which move into the fur and move through it when disturbed [41]. A subset of the bat flies in our dataset, 437 specimens collected at the Chucantí Nature Reserve, Darién Province, Panama [79], were used to investigate the association between ecomorphology of these bat flies and parasitism by Laboulbeniales. Statistical analysis was done using the “chisq” function in R [80]. 

## 3. Results

### 3.1. Association Web

The association web of the Laboulbeniales-infected bat flies can be found in Table 1 and Table 2 and in Figure 2. A total of 22 bat species, 22 bat fly species, and 9 Laboulbeniales species were included. A tenth node of Laboulbeniales included *Gloeandromyces* spp. not identified to species level. Note that *Trichobius* sp. nov. (Tri_new in Figure 2) and *Gloeandromyces* sp. nov. (Glo_nov in Figure 2) have not been formally described in the literature. The associations of each host species with their parasite species can be found in Appendix A. A detailed overview of all bat fly–Laboulbeniales associations thus far can be found in Appendix A.

### 3.2. Structural Specificity

We found very high host specificity at the community-level for the association of bat flies with their bat hosts (*H*_2_’ = 0.94). The community-level host specificity was also high for Laboulbeniales with their bat fly hosts (*H*_2_’ = 0.65), although more generalistic compared to the bat–bat fly association. The *d*’*_i_* values of each individual species with their associated host are presented in Table 1 and Table 2. Some species, such as *Trichobius joblingi*, were found on several host species but still have a high *d*’*_i_* value because a significant amount of specimens were found on a single host species (in the case of *T. joblingi*, on *Carollia perspicillata*).

### 3.3. Ecomorphology

Of the subset of 437 Neotropical bat flies collected at the Chucantí Nature Reserve, 69 were fur runners (of which 3 were infected by Laboulbeniales), 23 were fur swimmers (0 infected), and 345 were wing crawlers (27 infected). A chi-square test did not show a significant association between ecomorphology and parasitism by Laboulbeniales (χ^2^ = 2.4879, *p* = 0.2372).

## 4. Discussion

In this study, we analyzed associations among 22 bat species, 22 bat fly species, and 9 Laboulbeniales species from the Neotropics and Europe. Out of these, 15 associations between Laboulbeniales species and bat fly species had not been previously described (details in Appendix A). It is clear that the bat–bat fly–Laboulbeniales tripartite system is rich but still underexplored. Other tripartite surveys of Laboulbeniales on bat flies have been recently undertaken [79], but here we present the most up-to-date worldwide summary of bat–bat and fly–Laboulbeniales associations since Haelewaters et al. [51]. 

Our analysis of host specificity indicates that bat flies are very host-specific, whereas Laboulbeniales are less so. This creates a conundrum; how do ascospores of more generalist species of Laboulbeniales successfully transmit to a variety of bat fly species, when those bat flies are host-specific and stay most of their life on their bat host? Only pregnant female bat flies briefly leave their host bat to deposit a prepupa on the wall of the roosting environment, after which they need to re-colonize a new host individual for a blood meal. The pupal deposition area of *Trichobius* sp. is large, with the distance between observed pupae and the closest host bat roost (*Natalus stramineus*) ranging from 3.4 to 20.2 m [83]. Their depositing behavior—especially when distances are large—brings female bat flies in contact with several host bat individuals, increasing potential contacts with conspecific and heterospecific bat flies, and might be an additional reason that female bat flies have a higher frequency of Laboulbeniales infection [51]. Second, ascospores of Laboulbeniales are usually transmitted directly through interactions among arthropod hosts such as grooming and mating activities [46,49]. Because bats are more likely to be infested, more likely to carry heavier parasite loads, and more likely to harbor more species of bat fly in more permanent roosting structures (caves and tunnels, as opposed to leaf tents) [27], Laboulbeniales might have more opportunities to disperse to new host species in these roosts where many different bat fly species gather.

Another avenue of study could be to investigate the factors that determine why Laboulbeniales fungi are associated with different hosts. Are there any similarities in morphology or behavior among the bat fly hosts of a single species of Laboulbeniales, or do the bat hosts of those bat flies associated with a Laboulbeniales parasite species have a similar life history? Bat flies prefer specific body regions to settle on their host bats, and these preferences are associated with differences in bat fly morphology and behavior [38,41]. Our analysis of ecomorphology of bat flies versus Laboulbeniales infection as a whole saw no significant differences in infection prevalence for one or the other ecomorphological group. However, on a species-level, ascospores of Laboulbeniales might adhere easier to and develop better on bat fly body parts that are associated with specific bat fly ecomorphology—e.g., on the elongated posterior legs of fur runners. Laboulbeniales infections in other host groups such as beetles (Hexapoda: Coleoptera), corixids (Hexapoda: Hemiptera), and pill-millipedes (Diplopoda: Sphaerotheriida) can be position-specific, and some even exhibit sex-of-host specificity [49]. This makes it not unimaginable that bat fly ecomorphotype and the associated microhabitat on the bat host may affect the prevalence of Laboulbeniales infection as well, although the exact mechanics of these associations are as yet unstudied.

A few caveats must be considered. First, the geographic range of bats and their associates included in this study was limited to Costa Rica, Honduras, Mexico, Panama, Peru, and Trinidad (Neotropics), and Hungary and Romania (Europe). In future analyses, we should include data from more localities, including in African and Asian countries. Second, a few associations were represented by only a single specimen—in these cases, the single connection between parasite and host makes the parasite species highly “host-specific” while not necessarily being so. The use of *d*’*_i_* values for host specificity of species is a standard analysis tool and has been used to show specificity in, e.g., ticks and their hosts [16], bees and the plants they frequent [84], and scavengers and the carcasses they choose [85]. However, in our study, the interpretability of *d*’*_i_* values may be more limited, as *d*’*_i_* is dependent on host abundance; due to scarcity of the host and the parasite, even a parasite that only occurs on a single host species can get a value lower than 1. In a system of highly specialized parasites with low abundances such as the bat–bat fly–Laboulbeniales system, *d*’*_i_* values become distorted. More specimens would make the species-level host specificity measures more robust. Third, it should be noted that distributions of parasites among hosts might not only be determined by the availability of hosts; they are also influenced by abiotic factors such as temperature and humidity [78,86].

Parasites affect the topology and stability of food webs, and they influence ecosystem health [87,88]. The effects of Laboulbeniales on their hosts have not been well-studied. Laboratory bioassays have only been performed for one species; *Hesperomyces virescens* increases the mortality of native and invasive ladybirds [89]. But how do Laboulbeniales associated with bat flies affect their hosts? Resolving this question as well as if and how biotic and abiotic traits may affect the tritrophic interactions among bats, bat flies, and Laboulbeniales fungi, will continue to decipher the long-neglected bat-hyperparasitic system. Here, we studied and discussed associations and host specificity. All in all, parasites are a crucial factor that can shape food webs and species distribution patterns, but they are notoriously understudied. Recently, a global parasite conservation plan was put forward, proposing, among others, more concentrated efforts to describe their current distribution so that existing predictions can be validated and improved about how parasites and their hosts are affected by a changing world [3,5]. Studies such as this one aid ecologists, parasitologists, and conservation biologists to work towards a better global understanding of parasitism, and add to an increasing body of literature showing that we should not underestimate this omnipresent mode of life. 

## Figures and Tables

**Figure 1 jof-06-00361-f001:**
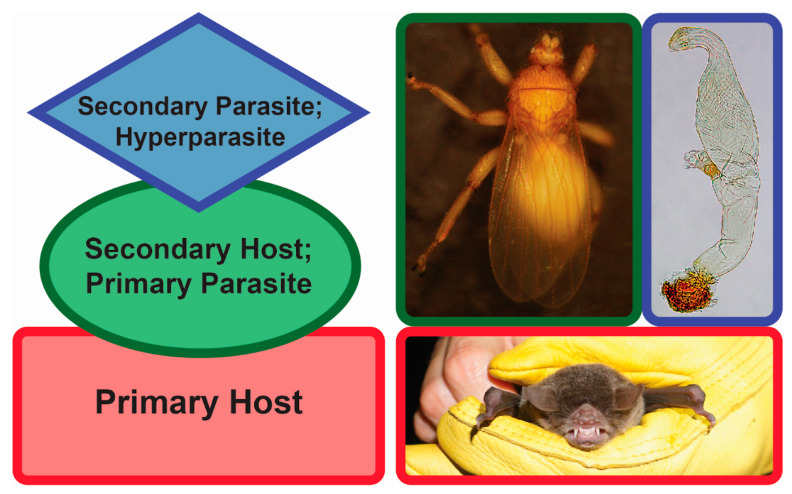
Hyperparasitism. Left, generalized diagram of hyperparasitism. Red, primary host (bat); green, secondary host/primary parasite (bat fly); blue, secondary parasite/hyperparasite (fungus). Right, *Pteronotus parnellii* (Mormoopidae), *Trichobius yunkeri* (Streblidae), *Gloeandromyces nycteribiidarum* (Laboulbeniales). Images not to scale. Photos: Danny Haelewaters, Thomas Hiller.

**Figure 2 jof-06-00361-f002:**
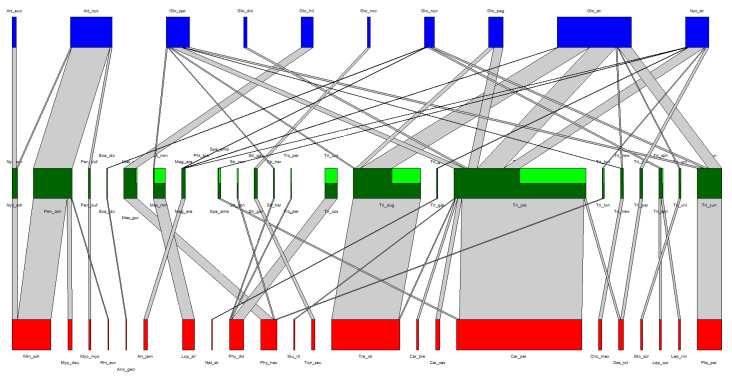
Quantitative bat–bat and fly–Laboulbeniales tripartite interaction network. Nodes (red, green, and blue) represent species, links (grey) represent species interactions. The width of the nodes and links corresponds to the quantitative frequency of surveyed species and the frequency of species interactions, respectively. Bat species nodes are in red, bat fly species nodes in green, and Laboulbeniales species nodes in blue. In the bat fly nodes, dark green represents those individuals of the bat fly species on which the Laboulbeniales was identified to at least genus level, whereas light green represents the individuals that were infected by Laboulbeniales but where the Laboulbeniales was not identified.

**Table 1 jof-06-00361-t001:** List of bat fly species and their associated bat hosts, numbers of collected specimens, and *d*’*_i_* values. Nycteribiid bat flies are shown in green, streblids in blue.

Bat Fly Species	N	*d*’*_i_*	Bat Species
*Nycteribia schmidlii*	5	0.47	*Miniopterus schreibersii*
*Penicillidia conspicua*	38	0.94	*Miniopterus schreibersii* *Myotis daubentonii* *Rhinolophus euryale*
*Penicillidia dufouri*	2	1	*Myotis myotis*
*Exastinion clovisi*	1	1	*Anoura geoffroyi*
*Mastroptera guimaraesi*	13	0.93	*Phyllostomus hastatus*
*Mastoptera minuta*	12	1	*Lophostoma silvicolum*
*Megistopoda aranea*	4	1	*Artibeus jamaicensis*
*Speisera ambigua*	2	0.12	*Carollia perspicillata*
*Strebla consocia*	1	0.42	*Phyllostomus hastatus*
*Strebla galindoi*	3	1	*Tonatia saurophila*
*Strebla hertigi*	1	0.45	*Phyllostomus discolor*
*Trichobioides perspicillatus*	1	0.45	*Phyllostomus discolor*
*Trichobius costalimai*	13	0.95	*Phyllostomus discolor*
*Trichobius dugesioides*	66	0.99	*Trachops cirrhosus*
*Trichobius galei*	1	1	*Natalus stramineus*
*Trichobius joblingi*	132	0.92	*Carollia brevicauda* *Carollia castanea* *Carollia perspicillata* *Desmodus rotundus* *Sturnira lilium* *Trachops cirrhosus*
*Trichobius longipes*	2	0.54	*Phyllostomus hastatus*
*Trichobius* sp. nov.	3	1	*Choeronycteris mexicana*
*Trichobius parasiticus*	3	0.88	*Desmodus rotundus*
*Trichobius sphaeronotus*	3	1	*Leptonycteris curasoae* *Leptonycteris nivalis*
*Trichobius uniformis*	2	1	*Glossophaga soricina*
*Trichobius yunkeri*	24	1	*Pteronotus parnellii*

**Table 2 jof-06-00361-t002:** List of Laboulbeniales and their association with bat fly species, numbers of collected specimens, and *d*’*_i_* values. Only Laboulbeniales identified to species level are included. Nycteribiid bat flies are shown in green, streblids in blue.

Laboulbeniales Species	N	*d*’*_i_*	Bat Fly Species
*Arthrorhynchus eucampsipodae*	4	0.94	*Nycteribia schmidlii*
*Arthrorhynchus nycteribiae*	41	0.97	*Nycteribia schmidlii* *Penicillidia dufouri* *Penicillidia conspicua*
*Gloeandromyces dickii*	3	0.19	*Trichobius joblingi*
*Gloeandromyces hilleri*	12	1	*Mastoptera guimaraesi*
*Gloeandromyces* sp. nov.	3	1	*Strebla galindoi*
*Gloeandromyces nycteribiidarum*	10	0.69	*Exastinion clovisi* *Megistopoda aranea* *Trichobius caecus* *Trichobius sphaeronotus* *Trichobius yunkeri*
*Gloeandromyces pageanus*	14	0.21	*Trichobius dugesioides* *Trichobius joblingi*
*Gloeandromyces streblae*	73	0.45	*Megistopoda aranea* *Trichobius dugesioides* *Trichobius joblingi* *Trichobius* sp. nov. *Trichobius uniformis* *Trichobius yunkeri*
*Nycteromyces streblidinus*	23	0.5	*Megistopoda aranea* *Speisera ambigua* *Trichobius galei* *Trichobius joblingi* *Trichobius longipes* *Trichobius parasiticus*

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
