# Peer review of "On the Fly: Tritrophic Associations of Bats, Bat Flies, and Fungi"

_jof, 2020, doi:10.3390/jof6040361_

Round 1

Reviewer 1 Report

The authors analyzed host specificity of a tri-trophic interaction among bats, flies and parasitic fungi. The manuscript is well written and is suitable for publication, after few corrections:

  • The collection site(s) (location, coordinates, etc.) should be specified at the beginning of materials and methods.
  • In which period of the year the specimens were collected? I think this is also a useful information for the reader.
  • Probably "Laboulbeniales", at line 112, shouldn't be written in italics. 

Author Response

Dear reviewer,

Thank you for taking the time to review our paper. With regards to your comments:

  • The collection sites and dates (year range and continents) have been added to the Materials and Methods at line 131. The full list of countries can be found in the discussion (line 255). The list of localities of the first associations can be found in Supplementary Table S3 and we will add an additional Supplementary Table S4 (newly mentioned in line 144 and 292) with the full list of localities with co-ordinates.
  • The periods of the year of collection range all over the entire year, as this pertains almost 12,000 bat flies collected over a period of 25 years. Therefore, the period of the year would be "the whole year", and not be very informative in this particular case.
  • "Laboulbeniales" on line 112 was in italics as it introduces a new concept at that point in the text; compare for example "Parasitism" on line 44 or "Bat flies" on line 99. After discussing it with the co-authors, we decided to leave it as-is. 

Kind regards,

Michiel de Groot

Reviewer 2 Report

Dear Authors,

Thank you for interesting article.

This study is clear that the bat–bat fly–Laboulbeniales tripartite system is rich.

I think that this article is accept level.

If there are problems, it has better that author draw the schematic diagram for “Materials and Methods”.

Best regards

Author Response

Dear reviewer,

Thank you for taking the time to review our paper and your comments. We are glad it was interesting to you. With regards to the schematic in the Materials and Methods, co-authors discussed this suggestion but we all feel that the methods are clear. In addition, both reviewers agreed that the methods are adequately described. As a result, we did not make a schematic diagram.

Kind regards,

Michiel de Groot